# Emerging Next-Generation Target for Cancer Immunotherapy Research: The Orphan Nuclear Receptor NR2F6

**DOI:** 10.3390/cancers13112600

**Published:** 2021-05-26

**Authors:** Victoria Klepsch, Kerstin Siegmund, Gottfried Baier

**Affiliations:** Institute for Translational Cell Genetics, Medical University Innsbruck, 6020 Innsbruck, Austria; kerstin.siegmund@i-med.ac.at (K.S.); gottfried.baier@i-med.ac.at (G.B.)

**Keywords:** tumor immunology, orphan nuclear receptor NR2F6, tumor-promoting function, alternative and druggable cancer immune checkpoint, transcriptional repressor of CD4^+^ and CD8^+^ effector but not regulatory T cell functions, NR2F6 inhibition as first-in-class cancer immunotherapy concept

## Abstract

**Simple Summary:**

The most successful strategies for solid cancer immunotherapy have centered on targeting the co-stimulatory and co-inhibitory T cell molecules that regulate T cell activation. Although immunotherapy that targets surface receptors such as CTLA-4 and/or PD-1 with recombinant antibodies has been a game changer for cancer treatment, a sizeable subset of patients still fail to respond to, and even fewer patients are cured by, these therapy regimens. Here, we discuss the unique potential of NR2F6 as an emerging target for cancer immunotherapy to significantly increase response rates of cancer patients and/or to extend treatment to a broader range of cancer types.

**Abstract:**

Additional therapeutic targets suitable for boosting anti-tumor effector responses have been found inside effector CD4^+^ and CD8^+^ T cells. It is likely that future treatment options will combine surface receptor and intracellular protein targets. Utilizing germline gene ablation as well as CRISPR/Cas9-mediated acute gene mutagenesis, the nuclear receptor NR2F6 (nuclear receptor subfamily 2 group F member 6, also called Ear-2) has been firmly characterized as such an intracellular immune checkpoint in effector T cells. Targeting this receptor appears to be a strategy for improving anti-tumor immunotherapy responses, especially in combination with CTLA-4 and PD-1. Current preclinical experimental knowledge firmly validates the immune checkpoint function of NR2F6 in murine tumor models, which provides a promising perspective for immunotherapy regimens in humans in the near future. While the clinical focus remains on the B7/CD28 family members, protein candidate targets such as NR2F6 are now being investigated in laboratories around the world and in R&D companies. Such an alternative therapeutic approach, if demonstrated to be successful, could supplement the existing therapeutic models and significantly increase response rates of cancer patients and/or expand the reach of immune therapy regimens to include a wider range of cancer entities. In this perspective review, the role of NR2F6 as an emerging and druggable target in immuno-oncology research will be discussed, with special emphasis on the unique potential of NR2F6 and its critical and non-redundant role in both immune and tumor cells.

## 1. Introduction—T Lymphocytes in Anti-Tumor Immunity

T lymphocytes are important mediators of cancer immunosurveillance. This hypothesis, which describes a role of immune cells in preventing the progression of neoplastic cells into cancer, was first formulated independently by Burnet and Thomas more than 60 years ago (reviewed in [1]). Since then, the importance of T cells in fighting tumors has been demonstrated in many studies with epidemiological and experimental evidence. For instance, mice lacking mature T lymphocytes are more susceptible to spontaneous tumor development and chemically induced tumors (reviewed in [2]). Furthermore, there is an increased incidence of malignancies in tissue transplant patients receiving immunosuppressive drugs. In addition, depletion of T lymphocytes, particularly CD8+ cytotoxic T lymphocytes (CTL), leads to impaired tumor rejection [3] and the control of dormant disseminated metastatic cells [4,5,6]. Consistent with these findings, an increased number of tumor-infiltrating T lymphocytes (TILs) were found to be associated with better survival prognosis [7]. TILs are a heterogeneous group with regard to the kind of T lymphocyte subsets they comprise, differentiation stage and activation status. CD8^+^ CTLs are considered the key effectors of anti-tumor immunity associated with favorable clinical outcomes since they have the capacity to directly kill tumor cells, for instance, by perforin and granzyme B release. The role of CD4^+^ TILs is rather ambiguous depending on the phenotype and, consequently, the quality of their response. Thus, Th1 cells contribute either directly, by killing MHC-class II^+^ tumors [8,9], or indirectly, through IFNγ secretion, to boosting the anti-tumor function of innate immune cells such as M1 tumor-associated macrophages [10]. Furthermore, CD4^+^ T lymphocytes increase MHC-class I and II expression on antigen-presenting cells (APC) and provide help, for instance, via IL-2 secretion, to CD8^+^ T cells to support the duration and magnitude of CTL responses. In contrast, CD4^+^ regulatory T cells (Treg) within the TME, which can suppress, for example, CTL or Th1 responses, are thought to interfere with effective anti-tumor immunity and therefore considered as indicators of a poor clinical outcome [11]. Of note, not only the mere number of cells of each T lymphocyte subtype but also their ratio in relation to the other subsets within the TME (e.g., CD8^+^/Treg cell ratio) is of great significance for predicting the survival outcome, which was shown to be the case in cervical cancer [12]. Thus, local Treg depletion is considered as one strategy to tip the balance among TILs towards effective anti-tumor immunity [13]. Another immune modulatory intervention, already used in clinics, aims to re-activate the anti-tumor response of exhausted T lymphocytes by checkpoint blockade therapy (ICB), which targets inhibitory receptors on T cells and thus releases the brake from these dysfunctional T cells. Moreover, the aim of vaccination strategies is to expand tumor antigen-specific T lymphocytes and increase their activation status and effector function. Furthermore, adoptive transfer of cells (ACT) such as chimeric antigen receptor (CAR)-engineered T cells, along with modifications in the T cell signaling cascade, provides a promising approach to harness T cell-mediated anti-tumor responses.

## 2. Current State of Tumor Immunology

### 2.1. Cancer Vaccination: A Strategy to Enhance T Lymphocyte-Mediated Anti-Tumor Immunity

One strategy to expand the pool of available tumor-specific T cells is therapeutic cancer vaccination. However, one hurdle in this approach is the identification of tumor-associated antigens (TAA) suitable for vaccination. Since tumor cells are derived from autologous cells, they are “self”, and T lymphocytes have been selected to be self-tolerant. Therefore, TAA-based vaccination approaches need to overcome self-tolerance mechanisms, by using, for instance, strong adjuvants, in order to induce robust CD4^+^ and CD8^+^ T cell immunity. However, this also harbors the risk of off-tumor on-target toxicity. Fortunately, in this context, tumors are often genetically unstable and thus show a high mutational burden that generates unique tumor-specific neoantigens (nAg), which are non-self-peptides without pre-existing central tolerance [14]. nAg vaccination approaches seem to be superior due to the reduced risk of inducing autoimmunity and due to a lower activation threshold of nAg-specific T lymphocytes, since there is no need to break immune tolerance of self-antigens for an efficient anti-tumor response (reviewed in [15]). However, because neoantigens frequently differ between patients, personalized vaccination protocols are needed, making them more expensive and less feasible for their wide use in clinical practice. Nevertheless, preclinical studies in several mouse models as well as recent studies have shown the effectiveness of nAgs cancer vaccination in therapy [16,17,18,19]. Still, a major drawback of many cancer vaccines tested thus far is their low efficiency in eliciting clinical responses, which might be largely due to the immune-suppressive milieu of the TME. One strategy to increase cancer vaccination success rates is to combine vaccination approaches with treatment regimens that target stimulatory and/or inhibitory co-receptors on TILs, by using agonists or blocking antibodies, respectively. Vice versa, expansion of tumor-specific T cells induced by vaccination has the potential to increase the clinical response rate to immune checkpoint inhibitors. These combination therapies have already been successfully tested in the first clinical studies [20,21,22].

Furthermore, vaccination strategies have also been used recently to expand adoptively transferred CAR-T cells in vivo, thereby improving their engraftment and persistence, which is known to be critical for their clinical efficacy. For example, Reinhard et al. showed in their so-called two-part “CARVac” strategy that a nanoparticulate mRNA vaccine that delivers the CAR antigen stimulates CAR-T cells, improves CAR-T cell engraftment and enhances tumor regression in a humanized mouse model [23].

### 2.2. Modulating T Lymphocyte Activation: Checkpoint Inhibitors, Co-Stimulatory Receptors and CAR-T

One major immune evasion mechanism of tumor cells is the upregulation of ligands of inhibitory receptors expressed by T lymphocytes, the so-called immune checkpoints (IC). Thereby, cancers hitchhike a molecular safeguard tactic intended to prevent self-tissue being attacked by immune cells. Thus, cancers generate a microenvironment that facilitates tumor growth. As mentioned above, targeting these inhibitory pathways to counteract the immune-suppressive activity of malignant cells has been an appealing strategy to treat cancer. Indeed, ICB has been demonstrated to be an effective approach for enhancing effector T cell activity in numerous preclinical models. More importantly, it has already been successfully applied in several types of cancer in the clinic (reviewed in [1,24]). Up until now, six antibody-based immune checkpoint inhibitors have been approved by the FDA for use in humans. Ipilimumab, the first inhibitor approved by the FDA in 2011, targets cytotoxic T lymphocyte antigen 4 (CTLA-4), while the other five interfere with the programmed cell death 1 (PD-1) axis, by either binding PD-1 on T lymphocytes or one of its ligands such as PD-L1. The importance of the discovery of inhibiting negative T lymphocyte regulators to treat cancer was recognized in 2018 by the award of the Nobel Prize in Physiology or Medicine to James P. Allison (CTLA-4) and Tasuku Honjo (PD-1).

CTLA-4, along with PD-1, is a member of the immunoglobulin superfamily and thus a transmembrane protein comprising an extracellular and a cytoplasmic domain, of which the latter is involved in intracellular signaling. The surface expression of these inhibitory receptors depends on the activation status of the T cell (reviewed in [1]). CTLA-4, which is expressed at a low basal level, is strongly upregulated early during the course of T cell activation. Interestingly, CD4^+^CD25^+^ Treg cells have been reported to constitutively express CTLA-4, and this has been shown to be essential for their direct and indirect immunosuppressive activity. PD-1 is transiently expressed during initial T cell activation. Strong and durable PD-1 upregulation occurs on T lymphocytes due to prolonged antigen exposure such as during chronic viral infection or in the TME, which leads to T lymphocyte exhaustion. Based on its expression profile and mode of action, CTLA-4 is mainly implicated in regulating early activation of naïve T lymphocytes, mostly taking place within lymphoid tissues. In contrast, PD-1 rather seems to control the continued activation and proliferation of differentiated effectors within peripheral tissues. These differences in the mode of action of CTLA-4 and PD-1 in T cell biology are also reflected by the outcome of ICB therapy that targets one or the other separately [25]. Anti-CTLA-4 ICB, for instance, supports the induction phase of anti-tumor T cell responses, while ICB that targets the PD-1/PD-L1 axis serves mainly to maintain the effector phase of anti-tumor T cell responses. Therefore, the mixture of anti-CTLA-4 and anti-PD-1/PD-L1 therapies not only showed synergistic effects in pre-clinical models but also increased their efficacy compared to monotherapies in clinical settings such as in patients with metastatic melanoma and advanced renal cell carcinoma [26,27]. However, parallel with clinical effectiveness, there was an increase in the frequency of immune-related adverse events (irAE) with combinatorial ICB regimens [28]. Overall, despite being shown to be an effective strategy to enhance anti-tumor T cell activity, the currently approved ICBs elicit quite low response rates within a range of 10–30% in most cancers. Thus, several other immune checkpoints such as lymphocyte activation gene-3 (LAG-3), T cell immunoglobulin and mucin-domain containing-3 (TIM-3), T cell immunoglobulin and ITIM domain (TIGIT) and, as recently described by our own group, the nuclear receptor orphan NR2F6 (Nuclear receptor subfamily 2, group F, member 6; alias Ear2 and COUP-TFIII) are under investigation to address the unmet medical need for improved cancer immunotherapy. Of note, these inhibitory receptors are highly and persistently expressed on exhausted T cells (reviewed in [29]). Hence, they are considered novel therapeutic targets to reactivate dysfunctional tumor-specific T lymphocytes, which is currently being tested in multiple clinical trials. However, thus far, ICBs other than the approved CTLA-4 and PD-1/PD-L1 treatments have not shown any substantial clinical benefit when applied as monotherapies [1,30].

An alternative approach to mobilize tumor-specific T lymphocytes against cancer comprises the activation of co-stimulatory pathways by agonists [27]. The major co-stimulatory pathway, often referred to as the second signal of T lymphocyte activation, involves CD28 engagement by CD80/86. Among other positive regulators are 4-1BB (CD137), OX40 (CD134), inducible T cell costimulator (ICOS) and glucocorticoid-induced tumor necrosis factor receptor (GITR). Many of these co-stimulatory receptors are upregulated upon TCR engagement and show increased expression on exhausted TILs, making them a promising target to improve checkpoint efficacy by “releasing the brakes and accelerating T cell activity”. Indeed, many clinical trials are currently testing agonistic antibodies against co-stimulatory molecules, either as single-agent studies or as combination therapies [31,32]. An active field is also research exploring strategies to avoid irAE due to “over-stimulation” of T lymphocytes. So-called bispecific antibodies that recognize a tumor-specific antigen (TSA) and the stimulatory receptor are employed to localize the T cell stimulus to the tumor. The bispecific antibodies TSAxCD28 and TSAx4-1BB, for instance, showed improved anti-tumor responses in studies using either mice or nun-human primates without increased irAE [30,33,34]. Furthermore, the signaling domains of co-stimulatory molecules such as CD28 and 4-1BB have been used alone or in combination to generate improved CAR-T cells (second and third generation), which are less prone to become exhausted and thus show enhanced persistence [35,36].

Personalized CAR-T cell therapy has now become the front line of cancer therapy (>500 clinical trials listed), starting with the FDA approval in 2017 of two CD19-CAR-T cell therapies for B cell lymphoma in adults and acute refractory leukemia in children, offering a great treatment option in the clinic. Technically, T cells derived from patient blood are engineered in vitro in such a way that they express artificial receptors that target a specific tumor antigen without the involvement of MHC [37]. Thus far, the FDA has approved CAR-T cell therapies for adult patients of certain types of non-Hodgkin lymphoma including aggressive, relapsed or refractory diffuse large B cell lymphoma, primary mediastinal B cell lymphoma, high-grade B cell lymphoma, transformed follicular lymphoma and mantle cell lymphoma, and for young adults and children with acute lymphoblastic leukemia that have not responded to other forms of treatment. Although CAR-T therapies are currently limited to hematological disorders with no efficacy in solid tumors, especially because of the suppression of T cell activity in the tumor microenvironment [38,39], future CAR-T therapy developments may represent a major advancement in personalized cancer treatment for both non-solid and solid tumors.

## 3. Beyond Current Immune Checkpoint Therapies

### 3.1. Intracellular Target NR2F6 in Both Immune Cells and Tumor Cells

The identification of alternate and potentially additive immune checkpoint candidates is intended to improve immunotherapies for a large number of cancer patients. Therefore, we focused on checkpoints located inside immune cells as suitable targets for future cancer drugs. We demonstrated, in recent years, the crucial T lymphocyte-intrinsic role of the orphan nuclear receptor NR2F6 as an intracellular checkpoint in fine-tuning adaptive immunity. NR2F6 induced an anti-inflammatory signal in the T cell compartment (Table 1). In agreement with this observation, Nr2f6-deficient mice spontaneously developed a late-onset autoimmune-type phenotype and were hyper-susceptible to the induction of neuroinflammation [40] (Figure 1). Furthermore, NR2F6 is ubiquitously expressed and found at rather low levels in resting T cells; however, it is highly inducible in effector (but not regulatory) T cells in an inflamed tumor microenvironment.

Recently, we showed that inhibition of NR2F6 gene function improves CD4^+^ and CD8^+^ T cell infiltration in addition to effector functions at the tumor site in different mouse tumor models [44,45]. In agreement with these observations, in all these pre-clinical models, a survival benefit and initiation of an anti-tumor immune response against both solid tumors and metastases were observed in Nr2f6-deficient mice [44,45]. Heterozygous Nr2f6^+/−^ mice similarly showed a strong benefit in terms of tumor growth suppression, demonstrating haploinsufficiency [50]. As clinical drug candidates eventually fail for a lack of efficacy, this dependence on less than 50% of NR2F6 function represents “great news” for inhibition efficacy during an envisioned therapy regimen of NR2F6-targeting drugs.

In our studies, validation of the biological importance of NR2F6 function in human T cells was carried out employing human NR2F6 knockdown T cell cultures. Furthermore, analyzing TILs in human NSCLC biopsy samples provides strong preclinical evidence that the upregulation of NR2F6 at the tumor site produces effector T cells incapable of achieving an adequate immune response against cancer. Most important is the strong synergistic effect of genetic ablation of NR2F6 in combination with an established blockade of surface checkpoints (PD-L1, CTLA-4). Importantly, these clear anti-tumor immune responses in the Nr2f6^−/−^ therapy groups did not show any signs of irAE that indicate local T cell activation [50]. These promising data form the basis for future clinical validation of NR2F6 targeting as a mechanistically independent and potentially synergistic option to improve the efficacy of immuno-oncological therapies.

Another immune cell type where NR2F6 has been shown to play an essential role is macrophages [51]. Of note, and conceivable for a nuclear receptor, it may regulate cytokine genes in an activating manner in human macrophages, contrary to an inhibiting manner in mouse macrophages [51], the latter being described in mouse effector T cells [43,44] (Figure 1). However, the mode of action of NR2F6 in T cells is the consistent transrepression in mice and humans.

Remarkably, beside its role in immune cells, NR2F6 expression also appears important in tumor cells, correlating with a faster tumor growth and worse patient overall survival outcomes (Table 1). Human studies firmly established an increased NR2F6 expression in various cancer types such as leukemia [41,56], colon carcinoma [39], cervical cancer [42], ovarian cancer [43,44], breast cancer [45,46,47,48,49,50,51,52,53,54,55,56,57], lung cancer [45,58] and hepatocellular cancer [38], suggesting that NR2F6 might be a prognostic marker. NR2F6 might directly regulate and elevate the survivability of cancerous colon cells via XIAP (X-Linked Inhibitor of Apoptosis) [39], whereas it supports proliferation and metastasis in hepatocellular carcinoma via TIP60 regulation [38], which is already known to promote mammary tumorigenesis [58], pleural mesothelioma malignancy [59] and prostate cancer [60] growth. In leukemia, NR2F6 increases long-term hematopoietic stem cells (LT-HSC) and prevents proliferative arrest associated with terminal differentiation [41] (Figure 1).

### 3.2. Inducible Immune Checkpoint at the Tumor Site May Boost a Localized Effector T Cell Response with Fewer Systemic Irae

As extensively discussed, immune checkpoint inhibitors have changed the treatment of cancer patients for the better. On the one hand, activated T cells are good and necessary for eradicating cancerous cells; on the other hand, irAE may occur and harm the patient receiving ICB because, by systemically blocking ICs, the delicate balance between activation and quiescence of the adaptive immune system is disrupted [61]. As a consequence, irAE arise that are quite serious. IrAE represent undesirable off-target immune and inflammatory events, thought to represent bystander effects systemically from activated T cells and augment the likelihood of autoimmune toxicities globally [62]. Nearly 80% of patients receiving monotherapy and up to 95% treated with combination therapy (PD-1 and CTLA-4) develop adverse clinical signs, most commonly rash, colitis, thyroiditis, hypophysitis, hepatitis, pneumotitis and arthritis. While these can occur any time during treatment, they develop most often within the first 3–4 months of therapy [63].

In this regard, our study, which targeted NR2F6 in mice in combination with PD-L1 treatment, found no exacerbated signs of irAE during a three-month follow-up period. When compared to wild-type mice similarly treated with PD-L1 ICB, no significant differences in immune cell infiltration, weight gain or colon length after anti-PD-L1 treatment were observed in mice with Nr2f6 deficiency [50]. This finding strongly suggests that NR2F6 might be an inducible and thus highly localized immune checkpoint at the tumor site. Of note, and in stark contrast, mice treated with a combination of anti-mouse PD-1 and CTLA-4 antibodies have been reported to develop severe diseases characterized by organ infiltration of immune cells [64].

This compartmentalization at the tumor site, per definition, decreases the chances of systemic side effects and thus should enable more precise therapy for patients during the envisioned NR2F6-targeted therapy regimen. Thus, therapies targeting NR2F6, once available, may help boost a localized effector T cell response at the tumor site with fewer systemic irAE. 

Furthermore, the observed NR2F6 upregulation in biopsies of patients with defined cancer entities [39,45,47,51,52,53,54,55,56,57,59] (Figure 1 and Table 1) might thus implicate an upregulation also of lymphatic NR2F6. This, however, still needs to be evaluated for patient stratification purposes.

### 3.3. Double Score Principle of NR2F6 Antagonists

NR2F6 has no known ligand thus far, and the search for its antagonists is intensive. If the search is crowned with success, it might represent a quantum jump in terms of increased survival of tumor patients in the future when used alone or in combination with other ICs. Of note, recent publications defined both closely related NRF2 family members NR2F2 [65,66] and NR2F1 [66] as druggable targets, indicating the feasibility of searching for a small-molecule antagonist targeting the immune checkpoint NR2F6, since it shares an evolutionary ligand binding domain (LBD) with its two family members NR2F1 and NR2F2 (Figure 2).

The regulatory mechanisms used by NR2F6 signaling, as far as they are known today, are depicted in Figure 3. Mechanistically, T cell-intrinsic NR2F6 directly suppresses the transcription of the transcription factors, namely, NFAT and AP-1, thereby repressing IL-2 and IFNγ secretion in activated T cells. Consistent with this ability of NR2F6 to reversibly disrupt NFAT/AP-1 DNA binding activities, several other nuclear receptors (NRs) such as GR and also PPAR, RXR, RAR and VDR have been reported to impair the ability of NFAT and/or AP-1 to transcribe their target genes in cells. In fact, this antagonism has been established to be the basis for the anti-inflammatory action of steroids [67,68].

Blocking NR2F6 thus can boost an anti-tumor immune response through direct transcriptional de-repression of key cytokine loci (e.g., Il2, Ifng in the T cell effector compartment [43,44]). The potential druggability of its LBD for a “small-molecule checkpoint blockade drug” therefore very likely offers a rational mechanistic basis for the targeted manipulation of NR2F6 in tumor-infiltrating T cells.

Furthermore, the inhibition of NR2F6 in tumor cells might also result in less survivability, proliferation and metastasis of the tumor cell [59,69]. Thus, and because cancer is a disease of malignant cells and immune cells, blocking NR2FR might kill two birds with one stone. Finally, an effect on cancer stem cells is possible as it is known for leukemia that NR2F6 increases long-term hematopoietic stem cells, while a blockade initiates terminal differentiation [41]. In fact, any rational drug design targeting NR2F6-mediated gene regulation would affect both effector T cells and tumor cells, simultaneously inducing tumor cell death and activating NR2F6 high-expressing TILs at the solid tumor site of, for instance, NSCLC. This double score principle, which targets NR2F6 in cancer patients, is intended to improve the effectiveness and expand the applicability of cancer immunotherapies and thus enable patients to survive longer.

## 4. Outlook

Why does the alternative and druggable NR2F6 cancer immune checkpoint appear to be important?

NR2F6 is a recently discovered immune checkpoint protein found at particularly high levels in T effector cells that have infiltrated tumors. In agreement with this observation, genetic NR2F6 blocking experiments have shown promising improvements in anti-tumor T cell responses, employing both human PBMC in vitro and pre-clinical cancer therapy models in vivo. Investigations of the inhibition of *Nr2f6* in mice that used an ex vivo CRISPR/Cas9-mediated gene ablation of *Nr2f6* in T cells prior to therapeutic ACT in conjunction with an approved PD-L1 or CTLA-4 ICB therapy improved this therapeutic anti-cancer activity [69]. As the future of the immuno-oncology therapy concept is positioned in combination therapies, NR2F6 might be an emerging next-generation target that combines intracellular as well as surface receptor pathways, thus improving T cell efficacy and therapeutic outcomes, and increasing the percentage of cancer patients who positively respond to treatment. In addition, a combinatorial approach of blocking NR2F6 signaling and initiating immunogenic cell death [70] by radiotherapy and/or chemotherapy (especially with adriamycin) would be preferable for a better disease outcome and future success.

The biological and clinical features of NR2F6 mentioned above establish it as a unique cancer therapeutic drug target. First, personalized adoptive therapy of genetically modified human T cells using CRISPR/Cas9 modification may induce exhaustion-resistant T cells at the tumor site, thus extending CAR-T therapy benefits to solid tumors such as NSCLC. Specifically, CAR-T cells combined with NR2F6 gene modification might thus represent an opportunity to extend the clinical efficacy of CAR-T cell immunotherapy to the treatment of advanced/metastatic NSCLC lung cancer in the future. Secondly, targeting the NR2F6 pathway using small-molecule inhibitors known to easily diffuse into the center of solid tumor masses may enable re-activation of exhausted T cells at the NSCLC tumor site. Performing experiments to identify ligands for the LBD of NR2F6 from tumor tissue, lipid species that co-immunoprecipitated with NR2F were detected using liquid chromatography coupled to mass spectrometry (LC-MS). Evidence of a selective ligand(s) extracted would support the hypothesis that endogenous NR2F6 ligands may exist and presumably modulate the active conformation to induce homo- and/or heterodimerization and recruitment of co-activators/co-repressors. This opens up the possibility of developing a first-in-class small-molecule drug that inhibits NR2F6 with oral bioavailability. This fascinating application potential may expand the success rate of immune-oncological therapies by offering the prospect of remission to late-stage metastatic cancer patients not responding to ICB and whose outcomes were previously invariably terminal.

As a note of caution, however, the target validation of NR2F6 is based on genetic evidence obtained from pre-clinical models only. Correlations between reduced NR2F6 expression levels and occurrence of autoimmune diseases such as systemic lupus erythematosus [71,72] have been reported. Nevertheless, it cannot be argued based simply on these reports that decreased NR2F6 protein levels in these individuals will also elicit enhanced anti-tumor immunity. To date, no malignant disease in humans is known that is directly related to a mutation or deletion of the NR2F6 locus.

## 5. Conclusions

Given the lack of human genetic data on the role of NR2F6 as a suppressor of effective cancer immunity and the current situation that not all therapeutics developed in murine tumor models achieve similar success in the clinic, the fate of NR2F6-based immuno-oncology therapy envisioned here ultimately depends on establishing functional antagonists for NR2F6 and their outcome in human clinical trials. Nevertheless, continued research on the orphan nuclear receptor NR2F6 represents a suitable path for drug discovery and might offer an innovative, mechanism-based, therapeutic strategy to augment the sensitivity of tumor-infiltrating T cells to tumor antigens in cancer patients. Therefore, the critical objective for now is to firmly validate this alternative cancer immune checkpoint, NR2F6, as a unique human cancer therapeutic candidate target of the T effector cell compartment for next-generation immune-oncology regimens.

## Figures and Tables

**Figure 1 cancers-13-02600-f001:**
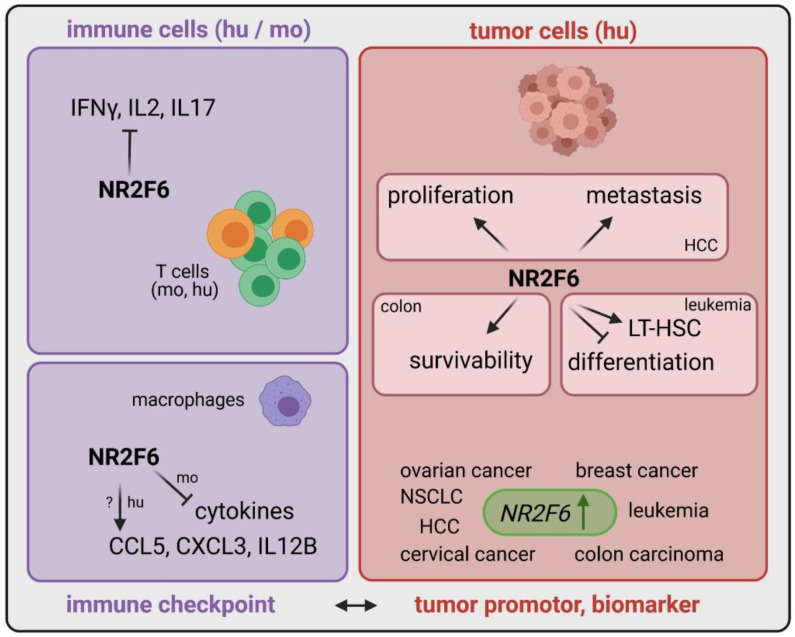
The dual role of NR2F6 in the immune system and tumor cells. In lymphocytes such as T cells (mouse and human), NR2F6 regulates the expression of cytokines such as IFNγ, IL2 and IL17 in an inhibiting manner reminiscent of an immune checkpoint. A similar mechanism of transrepression can be found in murine macrophages. However, apparently, NR2F6 may exert a functionally opposite transactivation role in human macrophages. Cytokine inhibition of immune cells reflects a tumor-promoting property. Moreover, and remarkably, upregulation in several human cancer cells such as ovarian, breast, cervical, colon, hepatocellular, non-small cell lung cancer and leukemia represents a prospective biomarker. In tumor cells, NR2F6 appears to be important for proliferation, metastasis, survivability of tumor cells and long-term hematopoietic stem cells (LT-HSC), whereas it inhibits differentiation processes in leukemic cells—all of which are tumor-promoting properties. In view of the dual pro-tumor activity of NR2F6 in immune cells and tumor cells, inhibition of NR2F6 offers the unique therapeutic potential to improve current treatment outcomes.

**Figure 2 cancers-13-02600-f002:**
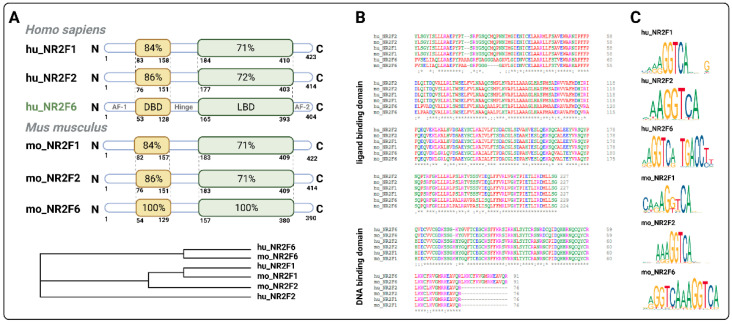
NR2F6 is the distantly related member of the NR2F family. (**A**) Schematic representation of the protein structure of NR2F1, NR2F2 and NR2F6 in human (hu) and mouse (mo). The different domains (AF-1–amino-terminal activation function (1) domain, DBD–DNA binding domain, hinge region, LBD–ligand binding domain, AF-2–activation function (2)) are represented by the percentage of amino acid identity among the receptors with respect to hu_NR2F6. The phylogram shown with cladogram branch length of the NR2F family defines the separation of NR2F6 from NR2F2 and NR2F1 as an early evolutionary event. Especially in the LBD, NR2F6 is significantly more divergent with homologies of 71% or 72% to the other family members, whereas NR2F1 and NR2F2 share the highest homology of 97%. (**B**) Multiple sequence alignment of LBD and DBD of mouse and human NR2F1, NR2F2 and NR2F6 (“*” = positions which have a single, fully conserved residue; “:” = conservation between groups of strongly similar properties; “.” = conservation between groups of weakly similar properties; color code: red = small; blue = acidic; magenta = basic–H; green = hydroxyl + sulfhydryl + amine + G; gray = unusual amino acids). (**C**) The GGTCA binding motif of the mouse and human NR2F family members. Programs used: Clustal Omega, Jaspar and pBLAST.

**Figure 3 cancers-13-02600-f003:**
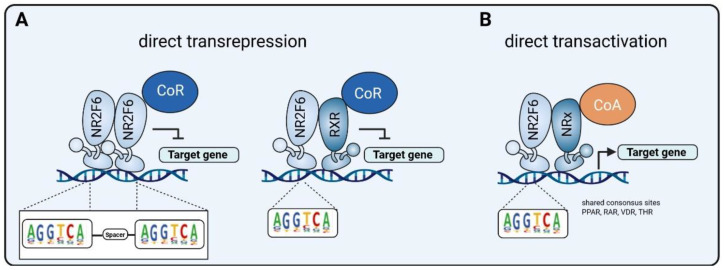
NR2F6 dimerization and mechanism of action. NR2F6 forms homo- and heterodimers (e.g., with RXRs) and acts, in a promoter context-dependent manner, either as a transcriptional repressor or an activator. Although natural ligands for NR2F6 are currently unknown, in the future, the potential druggability of its LBD for a “small-molecule checkpoint blockade drug” could provide a rational mechanistic basis for the targeted manipulation of NR2F6 in T cells. (**A**) By directly binding to GGTCA tandem motifs, i.e., as direct repeats separated by an appropriate spacer, and by forming complexes with a co-repressor (CoR), NR2F6 modulates target gene transrepression. Mechanistically, NR2F6 interferes with the transactivation of the target gene by other transcription factors and/or nuclear hormone receptors either through the sequestration of, e.g., RXR away from other DNA-bound protein–protein interaction complexes or through a direct occupation of their critical DNA-binding sites. (**B**) By direct binding as a heterodimer to a single GGTCA motif in close proximity to other DNA-bound nuclear receptors and by forming complexes with a co-activator (CoA), NR2F6 mediates transactivation of target gene expression. Adapted from Tang et al. [68].

**Table 1 cancers-13-02600-t001:** Role of NR2F6 in various cancer and cell types.

Cancer Type	Expression of NR2F6	Role of NR2F6	Ref
Leukemia	upregulated in patients	elevated population of LT-HSC	[41]
Hepatocellular carcinoma	upregulated in patients	NR2F6 induces proliferation and metastasis via circRHOT1 and TIP60	[38]
Colon carcinoma	upregulated in patients	Nr2f6 increases survivability via XIAP	[39]
Cervical cancer	upregulated in patients	correlation between metastasis, poor prognosis and NR2F6 expression	[42]
Ovarian cancer	upregulated in patients	DDA1 is induced by NR2F6 and predicts poor outcome	[43,44]
Breast cancer	upregulated in patients	n.d.	[45,46]
Lung cancer	upregulated in patients	MiR-142-3p inhibits proliferation, migration and invasion via NR2F6 inhibition	[47]
Effector T cells (mo and hu)	upregulated upon stimulation	transcriptional repressor directly antagonizing key cytokine gene loci	[40,48,49,50]
Macrophages (mo)	n.d.	transcriptional repressor of cytokines	[51]
Macrophages (hu)	n.d.	transcriptional activator of chemokines	[51]
Tumor cells	upregulated	important for proliferation, metastasis, survivability	[38,39,42,47,52]
Neurons (locus coeruleus)	n.d.	control of circadian clock	[53]
Hepatocytes (mo and hu)	upregulated	hepatic steatosis promoted by NR2F6	[54]
Kidney	n.d.	NR2F6 as a negative regulator of renin gene transcription	[55]

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
