# Peer review of "Emerging Next-Generation Target for Cancer Immunotherapy Research: The Orphan Nuclear Receptor NR2F6"

_cancers, 2021, doi:10.3390/cancers13112600_

Round 1

Reviewer 1 Report

Review is very well written and contains nice figures

The only thing that crossed my mind is that they might speculate a bit om the novelty of combining this immunotherapy approach with radiotherapy/chemotherapy in their outlook section?

Author Response

Point-by- point response to Reviewer 1:

Review is very well written and contains nice figures. The only thing that crossed my mind is that they might speculate a bit on the novelty of combining this immunotherapy approach with radiotherapy/chemotherapy in their outlook section?

Reply: We were remiss in not addressing this important point. We now added a sentence in the outlook section dealing with immunogenic cell death caused by radiotherapy/chemotherapy (page 10; line 384).

In addition, a combinatorial approach of blocking NR2F6 signaling and initiating immunogenic cell death [93] by radiotherapy and/or chemotherapy (especially with adriamycin) would be preferable for disease outcome and future success.”

Reviewer 2 Report

Klepsch et al., discussed the unique potential of NR2F6 as an emerging and druggable target for cancer immunotherapy which can significantly increase response rates of cancer patients. The authors provided a perspective on the usage of this protein target as an alternative therapeutic approach, if successfully demonstrated, could supplement/boost the existing immune checkpoint therapies. The authors further highlighted the critical and non-redundant role of NR2F6 in both tumor and immune cells.

Overall, the review is nicely written with lots of useful information. However, this review needs to be reorganized a bit. Introduction (1st and 2nd sections) are little lengthy and takes away the main focus of the review. I have some suggestions to make this review more interesting and appealing to a broader audience.

Major comments:

  1. Some of the redundant references can be removed. Most of the introductory text have been described in detail previously and can be cut short. Incorporate NR2F6 in the introduction. Starting with what is cancer immunotherapy and why most of the ICBs therapies are giving only 25-30% of positive outcome will be of great relevance. This is where NR2F6 will play an important role which can improve these outcomes to may be >50%.
  2. Dual role of NR2F6 in tumor and immune cells is not clear. A tabular comparison of the role of this molecule in immune and tumor cells with specific references will help readers understand better. Further adding a table describing positive and negative effects of NR2F6 in various diseases/variety of cancers, if any, will help the readers stay focused.

Author Response

Point-by- point response to Reviewer 2:

Klepsch et al., discussed the unique potential of NR2F6 as an emerging and druggable target for cancer immunotherapy which can significantly increase response rates of cancer patients. The authors provided a perspective on the usage of this protein target as an alternative therapeutic approach, if successfully demonstrated, could supplement/boost the existing immune checkpoint therapies. The authors further highlighted the critical and non-redundant role of NR2F6 in both tumor and immune cells.

Overall, the review is nicely written with lots of useful information. However, this review needs to be reorganized a bit. Introduction (1st and 2nd sections) are little lengthy and takes away the main focus of the review. I have some suggestions to make this review more interesting and appealing to a broader audience.

Reply: We appreciate that the reviewer considers our review as interesting to the field. Following his/her comments we edited the manuscript to make it even more appealing for interested readers.

Major comments:

  1. Some of the redundant references can be removed. Most of the introductory text have been described in detail previously and can be cut short. Incorporate NR2F6 in the introduction. Starting with what is cancer immunotherapy and why most of the ICBs therapies are giving only 25-30% of positive outcome will be of great relevance. This is where NR2F6 will play an important role which can improve these outcomes to may be >50%.

Reply: We thank the reviewer for carefully reading the manuscript and his/her valid comment. We shortened the introduction and removed redundant references. Also NR2F6 was incorporated in the introduction section beside other novel immune checkpoints (page4; line 158).

  1. Dual role of NR2F6 in tumor and immune cells is not clear. A tabular comparison of the role of this molecule in immune and tumor cells with specific references will help readers understand better. Further adding a table describing positive and negative effects of NR2F6 in various diseases/variety of cancers, if any, will help the readers stay focused.

Reply: Following the reviewer’s valid suggestion, we included a tabular comparison (Table 1) of the role of NR2F6 in immune and tumor cells, now highlighting its highly relevant role.

Reviewer 3 Report

In this review the authors discussed their finding on the Orphan Nuclear Receptor NR2F6 that they propose as a next generation target for cancer immunotherapy.

Based on their previous publications the authors described NR2F6 as Nuclear Receptor working as an intracellular T cell immune checkpoint. This is an interesting review, below are some comments that the authors may consider to implement their review:

1- structure of NR2F6: DNA binding domain, ligand binding domain: should be provided before

2- Unclear differences in macrophages in human versus mouse ? Any difference between human and mouse T cells?

3- as NR2F6 is propose as a target. It would be key to provide information on its distribution across tissues and immune and non-immune cell types.

4- The role of NR2F6 in other cells should be further reviewed: other immune cells, non immune cells, mechanisms in tumor cells : is NR2F6 also blocks NFAT/AP-1 DNA binding:?

5- Clarify if NR2F6 is upregulated on T cells at tumor site

6- irAE in mice:it would be important to compare anti-CTLA4 + anti-PD1 with NR2F6+anti-PD1 targeting

7- what are the strategies to identify the ligand(s) and agonists or antagonists? Any hypothesis based on similarity with other nuclear R?

8- need reference for activity on T cells NR2F6 in blocking NFAT/AP-1 DNA binding

9- figures at the end???

10- any SNP in NR2F6 and impact on cancer/pathologiy incidence

Author Response

Point-by- point response to Reviewer 3: In this review the authors discussed their finding on the Orphan Nuclear Receptor NR2F6 that they propose as a next generation target for cancer immunotherapy.

Based on their previous publications the authors described NR2F6 as Nuclear Receptor working as an intracellular T cell immune checkpoint. This is an interesting review, below are some comments that the authors may consider to implement their review:

  • structure of NR2F6: DNA binding domain, ligand binding domain: should be provided before

Reply: We thank the reviewer for carefully reading the manuscript and his/her valid comment. We now provide the structure in Figure 2 (exemplified in hu_NR2F6), and also refer to the literature (Kleiter 2012), where the structure is already nicely described.

  • Unclear differences in macrophages in human versus mouse? Any difference between human and mouse T cells?

Reply: With regard to T cells the mode of action of NR2F6 is consistent between human and mouse: transrepression. (page 7, line 258).

  • as NR2F6 is propose as a target. It would be key to provide information on its distribution across tissues and immune and non-immune cell types.

Reply: We thank the reviewer for the valid suggestion and in the revised MS properly discuss the expression of NR2F6 in the text and in the newly included Table 1.

  • The role of NR2F6 in other cells should be further reviewed: other immune cells, non immune cells, mechanisms in tumor cells : is NR2F6 also blocks NFAT/AP-1 DNA binding:?

Reply: Following the reviewer’s valid suggestion, we extended this part of NR2F6 in other cells, now including a table in the revised MS.

  • Clarify if NR2F6 is upregulated on T cells at tumor site

Reply: We already mentioned that NR2F6 is upregulated in T cells at the tumor site ( “… strong preclinical evidence that the upregulation of NR2F6 at the tumor site reproduces effector T cells incapable of achieving an adequate immune response against cancer.” (page 6; line 245)

  • irAE in mice:it would be important to compare anti-CTLA4 + anti-PD1 with NR2F6+anti-PD1 targeting

Reply: We thank the reviewer for carefully reading the manuscript and his/her valid comment. The combination of NR2f6+anti-PD-L1 targeting has been performed in our Nature Communication publication from 2018. A recent publication (2021) tested the combination of PD-1 and CTLA-4, were mice developed more substantial disease. Please find the according paragraph in the revised MS.

“In this regard, our study, which targeted NR2F6 in mice in combination with PD-L1 treatment, found no exacerbated signs of irAE during a three-month follow-up period. When compared to wild-type mice similarly treated with PD-L1 ICB, no significant differences in immune cell infiltration, weight gain, or colon length after anti-PD-L1 treatment were observed in mice with Nr2f6-deficiency [61]. This finding strongly suggests that NR2F6 might be an inducible and thus highly localized immune checkpoint at the tumor site. Of note, and in stark contrast, mice treated with a combination of anti-mouse PD-1 and CTLA-4 antibodies have been reported to develop severe diseases characterized by organ infiltration of immune cells [85] (page 7; line 287)

  • what are the strategies to identify the ligand(s) and agonists or antagonists? Any hypothesis based on similarity with other nuclear R?

Reply: In the revised MS we added a section on how a possible deorphanisation of NR2F6 could be performed.

“Performing experiments to identify ligands for the LBD of NR2F6 from tumor tissue, lipid species that co-immunoprecipitated with NR2F were detected using liquid chromatography coupled to mass spectrometry (LC-MS). Evidence of selective ligand(s) extracted would support the hypothesis that endogenous NR2F6 ligands may exist and presumably modulate the active conformation to induce homo- and/or hetero-dimerization and recruitment of co-activators/co-repressors. This opens up the possibility of developing a first-in-class small molecule drug that inhibits NR2F6 with oral bioavailability. This fascinating application potential may expand the success rate of immune-oncological therapies by offering the prospect of remission to late-stage metastatic cancer patients not responding to ICB and whose outcomes were previously invariably terminal.” (page 10; line 396)

  • need reference for activity on T cells NR2F6 in blocking NFAT/AP-1 DNA binding

Reply: We thank the reviewer for carefully reading the manuscript and his/her valid comment. The References for the activity on T cells are already in the MS.

“Mechanistically speaking, NR2F6 acts as an essential signaling intermediate; it sets the threshold of T cell effector functions by acting as a transcription repressor that directly antagonizes the DNA accessibility of activation-induced nuclear factor of activated T cells (NFAT) and activating protein 1 (AP-1) transcription factors at crucial cytokine gene loci such as Il2, Il17 and Ifng [58–60](page 5; line 211)

Please find again the according References:

  1. Hermann-Kleiter, N.; Gruber, T.; Lutz-Nicoladoni, C.; Thuille, N.; Fresser, F.; Labi, V.; Schiefermeier, N.; Warnecke, M.; Huber, L.; Villunger, A.; et al. The Nuclear Orphan Receptor NR2F6 Suppresses Lymphocyte Activation and T Helper 17-Dependent Autoimmunity. Immunity 2008, 29, 205–216, doi:10.1016/j.immuni.2008.06.008.
  2. Hermann-Kleiter, N.; Meisel, M.; Fresser, F.; Thuille, N.; Müller, M.; Roth, L.; Katopodis, A.; Baier, G. Nuclear orphan receptor NR2F6 directly antagonizes NFAT and RORγt binding to the Il17a promoter. J. Autoimmun. 2012, 39, 428–440, doi:10.1016/j.jaut.2012.07.007.
  3. Hermann-Kleiter, N.; Klepsch, V.; Wallner, S.; Siegmund, K.; Klepsch, S.; Tuzlak, S.; Villunger, A.; Kaminski, S.; Pfeifhofer-Obermair, C.; Gruber, T.; et al. The Nuclear Orphan Receptor NR2F6 Is a Central Checkpoint for Cancer Immune Surveillance. Cell Rep. 2015, 12, 2072–2085, doi:10.1016/j.celrep.2015.08.035.
  • figures at the end???

Reply: We have now implemented the figures in the text. We thought this would be done by the Journal just before publication.

  • any SNP in NR2F6 and impact on cancer/pathologiy incidence

Reply: We performed a literature search, which revealed that no SNP in NR2F6 in the context of cancer is known. However, there are NR2F6-SNPs described for Lupus, as mentioned in the MS:

“Correlations between reduced NR2F6 expression levels and occurrence of autoimmune diseases such as systemic lupus erythematosus [94, 95] have been reported. Nevertheless, it cannot be argued based simply on these reports that decreased NR2F6 protein levels in these individuals also will elicit enhanced anti-tumor immunity. To date, no malignant disease in humans is known that is directly related to a mutation or deletion of the NR2F6 locus”. (page 10; line 402)